# Semi-Supervised Siamese Network for Identifying Bad Data in Medical Imaging Datasets

**Niamh Belton**[1,2]                                   NIAMH.BELTON@UCDCONNECT.IE

**Aonghus Lawlor** [3,4]                               AONGHUS.LAWLOR@UCD.IE

**Kathleen M. Curran** [1,2]                           KATHLEEN.CURRAN@UCD.IE

[1]*Science Foundation Ireland Centre for Research Training in Machine Learning*

[2]*School of Medicine,* [3]*School of Computer Science, University College Dublin*

[4]*Insight Centre for Data Analytics, University College Dublin, Dublin, Ireland*

## Abstract

Noisy data present in medical imaging datasets can often aid the development of robust models that are equipped to handle real-world data. However, if the bad data contains insufficient anatomical information, it can have a severe negative effect on the model's performance. We propose a novel methodology using a semi-supervised Siamese network to identify bad data. This method requires only a small pool of 'reference' medical images to be reviewed by a non-expert human to ensure the major anatomical structures are present in the Field of View. The model trains on this reference set and identifies bad data by using the Siamese network to compute the distance between the reference set and all other medical images in the dataset. This methodology achieves an Area Under the Curve (AUC) of 0.989 for identifying bad data. Code will be available at https://git.io/JYFuV.

**Keywords:** Siamese Network, Semi-Supervised, Noisy Data, Bad Data, Contrastive Loss.

## 1. Introduction

Machine Learning (ML) models are developed for various medical imaging tasks. While it is important for ML models to be robust to noisy data, training such models on 'bad data' with irrelevant or no anatomical structures present in the Field of View (FOV) can harm the model's performance. In this paper, we present a semi-supervised Siamese network that can be implemented as a pre-processing step before medical image analysis to remove bad data. Siamese networks are also known as one-shot classifiers, meaning they can train on a small number of examples of a class and make predictions about unknown class distributions (Koch et al., 2015). We leverage this property to develop a bad data detector that trains on a small sample of good data and identifies many different types of bad data that have not been seen by the model during the training process.

In ML, there are many existing solutions for detecting anomalous data. These include semi-supervised One-Class methods and unsupervised methods such as Isolation Forest (Liu et al., 2008) and Autoencoders (Pang et al., 2021). We demonstrate that our proposed method outperforms Isolation Forests, while training on only a fraction of the data that other methods require. Our method, therefore, reduces training time and the amount of labelling required in comparison to semi-supervised methods.

## 2. Method

For this analysis, we used sagittal plane knee MRIs from the open-source MRNet dataset (Bien et al., 2018). Figure 1(i-ii) illustrates our proposed method. We curated a reference set of MRIs by choosing a random sample of 20 MRIs from the dataset. The labelling process involved a non-expert reviewing the MRIs in the reference set to ensure that the major knee structures were visible in the FOV. Thus, the model trains on only 20 MRIs from one class and it does not require labels for bad data. We choose a size of 20 to minimise the time required for the labelling process. In our experiments, we found that increasing the size of the reference set did not significantly impact the performance. Future work will investigate generalised methods of selecting the optimal reference set.

In each iteration of the training process, two reference MRIs were input into separate models that have an AlexNet architecture and shared weights. The model weights were initialised with weights trained on the ImageNet dataset and all weights were then subsequently trained. The model creates a 1-dimensional feature vector for each of the input MRIs and calculates the Euclidean Distance (ED) between the pair of feature vectors. The model was trained using Contrastive loss which penalises the model for outputting large EDs when comparing MRIs from the reference set. The model was trained with a batch size of one and for a period of six epochs.

Once the model was trained, all MRIs in the test set were input into the model separately. The ED between their output feature vectors and each feature vector in the reference set was calculated and averaged for each MRI. This assigns each MRI a Mean Euclidean Distance (MED) score. Large MED values indicate that the input MRI is dissimilar to the reference set and therefore, the input MRI is likely to be bad data.

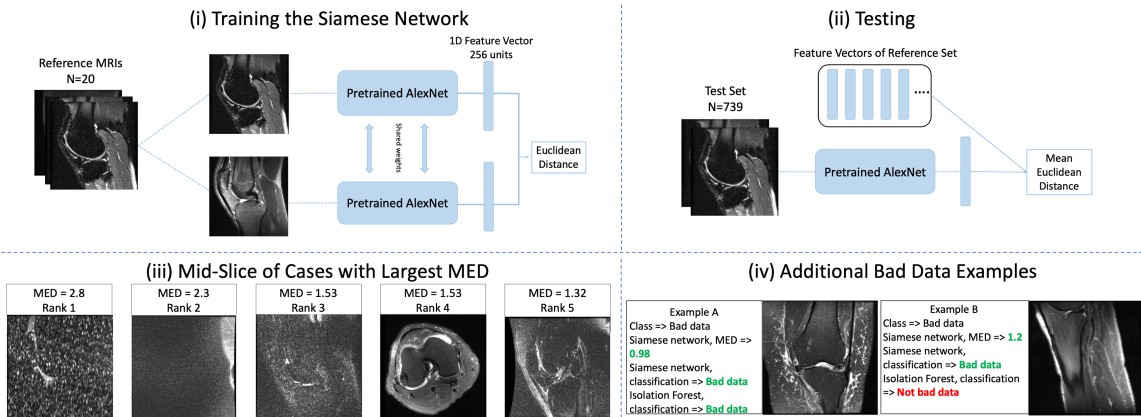

Figure 1: (i) The training process. (ii) The process for calculating the Mean Euclidean Distance (MED) for each MRI. (iii) The mid-slice of MRIs with highest MED values. (iv) Additional bad data examples.

## 3. Results

Figure 1(iii) presents the MRIs with the highest Mean Euclidean Distance (MED). This result demonstrates the Siamese network's ability to identify many different types of bad data, none of which the model was trained on. For example, it has identified data that has no visible anatomical information throughout the MRI (rank one and two) and it also identified an MRI from the axial plane that is mistakenly included in the sagittal plane data (rank four). Although rank three and five do show the relevant anatomical information, they are of poor quality and they could be considered for removal from the data set.

Table 1 shows the results for our method and a baseline method, Isolation Forest (IF) on a test set of 739 MRIs. There are seven cases of bad data in the test set. Baseline implementation and labelling details of the test set are available in the Github repository. The largest ED between pairs of reference MRIs was used as the MED threshold to determine what is classified as bad data. Although the Siamese network was trained on only 4% of the baseline method's training data, it showed a substantial performance improvement. However, it can be noted that IF is less computationally expensive. Figure 1(iv)(A) shows an MRI that we consider to be bad data given that it is acquired from the coronal plane and therefore, it is wrongfully included in the sagittal plane data. This MRI appears highly similar to the MRIs in the reference set and a non-expert human may find it difficult to make a distinction. Both the Siamese network and IF classified this as bad data. Figure 1(iv)(B) shows an MRI where the important anatomical information is mostly outside the FOV. IF misclassified this example, while the Siamese network accurately classified it as bad data.

Table 1: Model Performance Comparison

| Model | AUC | Sensitivity | Specificity |
|---|---|---|---|
| Siamese Network (proposed) | 0.989 | 100% | 89% |
| Isolation Forest | 0.802 | 71% | 92% |

## 4. Discussion and Conclusion

As part of our analysis, we assessed the sensitivity of the model's performance to the selection of reference MRIs. We ran multiple experiments with randomly sampled reference sets. All experiments had an AUC in the interval (0.983, 0.989). In this work, we have presented a methodology that achieves good performance, identifies a wide variety of bad data and requires only a fraction of the training data that previous methods require. This work has the potential to become a standard pre-processing technique for medical imaging analysis. In future work, we will test our technique on larger publicly available datasets and compare the method to additional baseline methods.

## Acknowledgments

This work was funded by Science Foundation Ireland through the SFI Centre for Research Training in Machine Learning (18/CRT/6183). This work is supported by the Insight Centre for Data Analytics under Grant Number SFI/12/RC/2289_P2.

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
