# OpenReview forum: "Semi-Supervised Siamese Network for Identifying Bad Data in Medical Imaging Datasets"
_MIDL.io/2021/Conference/Short — MIDL 2021 Poster_

### Official Review · Reviewer_39fJ · 2021-04-26

**Confidence:** 3
**Final Rating:** 1

**Summary:**

The authors propose a Siamese network to identify 'bad images' in medical imaging datasets (here, in knee MRI data). They demonstrate that their network can learn from a modest amount of examples to automatically identify which images deviate from having relevant anatomical structures. They validate their method on a publicly available knee-MRI dataset (MRNet) and demonstate improved performance over the Isolation Forests baseline.

**Strengths:**

1. The application of Siamese networks tasked at identifying anomalous images automatically is an interesting proposition, although the methodology adopted is not particularly novel in formulation. This could have important consequences for curating and standardising medical imaging databases in the future.

2. The proposed method outperforms the state-of-the-art effectively by learning from a smaller pool of samples.

**Weaknesses:**

1. My main concern is that the experimental comparisons are a bit weak and seemingly incomplete for drawing the strong conclusions that the abstract mentions and subsequent claims of clinical utility:

a) For example, section 3 reads as if the data was tested on mainly sagittal plane references. However, for claiming that the method does equally well on the dataset, I think it would be meaningful to see if the same network can be trained on a different plane and provide equally competitive performance

b) The authors do not mention explicitly what they mean by 'relevant anatomical information' in the FOV at annotation time.

c) The comparisons in Table 1 only indicate the AUC metric, without mention of the sensitivity, specificity etc. However, this can be pretty unreliable if the dataset has considerable imbalance within the classes. Moreover, I think its important to note precisely how many examples are considered as 'bad' within the dataset at test time.

d) As a thought experiment, if one were to flip (say horizontally) a relevant image at test time, it seems as if this would result in a considerably large Euclidean distance if the reference set does not contain such examples. Perhaps I am missing something, but the authors should probably comment on what prevents this from being labeled as a 'bad' example since they do not use any augemented data at training?

e) Since the authors chose to indicate the ranking of the examples according to the ED, an important metric to mention would be the False Positive rate. Is the inherent ranking order meaningful?

2. My second concern is about how the authors determined the training schedule, and particularly the early stopping at six epochs. Do they use a validation set to determine this? If so, I believe its worth mentioning within the main paper.

**Deanonymize Review:**

no

**Detailed Comments:**

Apart from the points detailed in weaknesses:

a) What are the relative capacities of the model vs the baseline (Isolation Forest), i.e. number of tunable hyperparameters?

b) What is the relative class ratio (bad vs good example)? I understand that the readers are directed to the Github repository, but my sense is that this information is relevant to the main argument and hence should be in the paper.

c) Why did the authors chose to train on exactly 20 images? How were these chosen. Does the performance degrade upon decreasing this number? By how much?

d) It would be meaningful to have a measure of variation over the AUC perhaps over multiple runs choosing a different set of images for training. In addition, consider expanding Table 1 to include metrics such as AU-PRC, sensitivity, specificity etc.

**Justification Of The Rating:**

I would categorise the main contribution of the paper as a novel application (detection of `bad examples) of a generic existing methodology (Siamese-networks and Alex Net). With that said, even for a short paper, the experimental section is not thorough, with several points major points unexplained (detailed in weaknesses and comments).

**Paper Type:**

methodological development

**Special Issue:**

no

---

### Official Review · Reviewer_RR4m · 2021-05-01

**Confidence:** 3
**Final Rating:** 4

**Summary:**

In this paper, the authors present a network which can be trained with only a few samples to remove bad or anomalous data. This task would serve as a pre-processing event prior to training another model for a higher-level task at hand.
In particular, the authors curate 20 reference samples from the MRNet dataset. A network pretrained on the ImageNet dataset is then finetuned by using contrastive loss in order to learn a 256-D  vector for each of the input reference images.
When evaluated on the test data set, all samples which produce a vector which is at a high Mean Euclidean Distance (MED) from the reference set is deemed as a bad sample. Using this strategy, the authors are able to operate at an AUC of 0.989 on the test dataset.

**Strengths:**

- The authors show that their methods achieves superior performance compared to the baseline method 'Isolation Forests', while training on only a fraction of the data. Thus, their method reduces training time.
- The authors run multiple experiments which produced an AUC between 0.983 and 0.989
- The authors make the corresponding code publicly available.

**Weaknesses:**

- I could not locate the threshold used on the MED metric, for defining a good and bad sample. (My understanding is that in order to evaluate true positives, false positives and false negatives, one must define a threshold on the mean euclidean distance. If for the queried sample, the MED is above this threshold it would be considered a bad sample, otherwise not. )

**Deanonymize Review:**

yes

**Detailed Comments:**

- "The model was trained with batch sizes of one and six epochs" could be reframed to "The model was trained with a batch size of 1 and for a period of 6 epochs".
 - Including a study where post the removal of bad samples using the strategy suggested in the paper, a model could train better on a higher level task would really drive the point of the paper.
- How does this method fare in comparison to [Pang et al, 2021](https://arxiv.org/pdf/2007.02500.pdf)?

**Justification Of The Rating:**

In this work, the authors provide a simple preprocessing strategy for identifying bad samples. This is a neat solution which could easily be introduced prior to training another model on higher level tasks.
I enjoyed the writing and reading about the motivation behind the paper. The theory and the experiments look solid. For this, I would rate this submission as a 'Strong Accept'.

**Paper Type:**

methodological development

**Special Issue:**

yes

---

### Meta-Review · Program_Chairs · 2021-05-10

**Recommendation:** Accept (Poster)
**Confidence:** 4

**Metareview:**

While not entirely novel as one of the reviewers indicates, the paper provides a nice solution to image dataset cleaning based on Siamese nets and contrastive loss, with good results. I recommend acceptance but would ask the authors to revise their work based on reviewers' comments. Limitations should be stated clearly.

---

### Decision · Program_Chairs · 2021-05-11

Accept (Poster)